# Preparation of SiO_2_@Au Nanoparticle Photonic Crystal Array as Surface-Enhanced Raman Scattering (SERS) Substrate

**DOI:** 10.3390/nano13152156

**Published:** 2023-07-25

**Authors:** Dingyu Song, Tianxing Wang, Lin Zhuang

**Affiliations:** Institute for Solar Energy Systems, Guangdong Provincial Key Laboratory of Photovoltaics Technologies, School of Physics, Sun Yat-sen University, Guangzhou 510006, China; songdy3@mail2.sysu.edu.cn (D.S.); wangtx7@mail2.sysu.edu.cn (T.W.)

**Keywords:** SERS, photonic crystal, silica nanoparticle, spin-coating, localized surface plasmon resonance effect

## Abstract

Surface-enhanced Raman scattering technology plays a prominent role in spectroscopy. By introducing plasmonic metals and photonic crystals as a substrate, SERS signals can achieve further enhancement. However, the conventional doping preparation methods of these SERS substrates are insufficient in terms of metal-loading capacity and the coupling strength between plasmonic metals and photonic crystals, both of which reduce the SERS activity and reproducibility of SERS substrates. In this work, we report an approach combining spin-coating, surface modification, and in situ reduction methods. Using this approach, a photonic crystal array of SiO_2_@Au core–shell structure nanoparticles was prepared as a SERS substrate (SiO_2_@Au NP array). To study the SERS properties of these substrates, Rhodamine 6G was employed as the probe molecule. Compared with a Au-SiO_2_ NP array prepared using doping methods, the SiO_2_@Au NP array presented better SERS properties, and it reproduced the SERS spectra after one month. The detection limit of the Rhodamine 6G on SiO_2_@Au NP array reached 1 × 10^−8^ mol/L; furthermore, the relative standard deviation (9.82%) of reproducibility and the enhancement factor (1.51 × 10^6^) were evaluated. Our approach provides a new potential option for the preparation of SERS substrates and offers a potential advantage in trace contaminant detection, and nondestructive testing.

## 1. Introduction

As a new trace signal detection technology, SERS spectra play a significant role in analytical techniques [1,2,3]. Compared with Raman spectra, SERS offers a huge signal enhancement, constituting its major feature, and the fingerprint nature of its signal is also a unique advantage [4]. These characteristics make SERS applicable in food testing [5,6,7,8], environmental pollutant detection [9,10], and nondestructive testing [11,12]. The SERS enhancement mechanism was derived from a chemical mechanism (CM) and an electromagnetic mechanism (EM), among which the EM effect was considered to be the dominant enhancement mechanism [13]. The causes of EM include hot spots and the localized surface plasmon resonance (LSPR) effect, whose combined action can increase SERS activity with an enhancement factor of up to 10^8^ in the best case [14]. CMs are generated by a charge transfer between the conduction band of a metal substrate and absorbed molecules, which enhances the polarizability of absorbed molecules and thus their Raman scattering cross-section [13]. This charge transfer can occur in both directions between the absorbed molecules and the metal substrate [15]. Gold (Au), silver (Ag), and other noble metal nanoparticles (NPs) have a strong LSPR effect in the visible to near infrared region and have been widely used in the preparation of SERS substrates [16,17]. Although Ag has high SERS activity, it is easy to oxidize and unstable, so Au, whose SERS activity and stability both meet the requirements in this regard, was adopted as a suitable material. The LSPR effect is particularly strong when the physical structure around metal NPs is a nanoscale periodic structure that is consistent with the structure of photonic crystals, and the photonic band gap (PBG) structure of photonic crystals can further enhance the effect of SERS. Silica NPs are often chosen as the basic unit of photonic crystals because they are stable, environmentally friendly, and neutral to redox reactions [18]. By combining the LSPR effect and a photonic crystal structure, a SERS substrate with outstanding performance can be obtained.

The doping method is a commonly used conventional method for preparing photonic crystal substrates. In this case, a limited doping amount of metal NPs is used. This weakens the SERS activity and reproducibility of substrates, posing a challenge in application [19]. To solve these problems, a great deal of research has been dedicated to improving the preparation methods of SERS substrates. Marko et al. [20] proposed the use of silver-coated porous silicon photonic crystals as SERS substrates. The substrates showed high SERS activity with a detection limit of 1 × 10^−7^ M for Rhodamine 6G (R6G). Wang et al. [21] developed a magnetically photonic chain-loading system substrate that can be modulated dynamically using an external magnetic field. The substrate realized rapid signal enhancement and SERS enhancement factor tuning through the loading of magnetically photonic nanochains of Fe_3_O_4_@SiO_2_ magnetic NPs. Zhong et al. [22] obtained silver NP-decorated porous silicon photonic crystals using an immersion plating solution, whose detection limit of picric acid reached 1 × 10^−8^ M. Zhang et al. [23] presented a facile method of reproducibly producing a monolayer photonic crystal substrate from SiO_2_@nAg NPs which were synthesized via the reduction of Ag NPs on SiO_2_ NPs. The test results with respect to p-aminothiophenol showed that the method yielded an enhancement of Raman scattering by a factor of 2.2 × 10^7^. Sansone et al. [24] proposed a substrate based on a metal–dielectric photonic crystal which was assembled using polystyrene NPs doped with Au NPs. When the laser excitation wavelength overlapped with the PBG of the substrate, the detection limit of R6G was ≤1 × 10^−6^ M (mol/L). Tuyen et al. [25] developed a SERS substrate composed of a three-dimensional polymeric photonic crystal structure with Au nanorods decorating the top layer. The inverse opal photonic crystal frame employs the constructive interference of a backward Raman signal and produces a strong and uniform SERS output.

In order to improve the SERS properties of substrates, we propose a method that differs from the conventional doping method. This method is based on spin-coating, surface modification, and in situ reduction methods and can be used to successfully increase the doping amount of metal NPs for a substrate. Through this process, a SiO_2_@Au NP array was prepared which served as the SERS substrate. Using R6G as the probe molecule to study the SERS characteristic of the substrate, the results indicated that the SiO_2_@Au NP substrate presented better performance than the Au-SiO_2_ NP substrate prepared using the doping method. With an enhancement factor of 1.51 × 10^6^, the detection limit of R6G reached 1 × 10^−8^ M, and the average of the RSD of reproducibility was 9.82%.

## 2. Materials and Methods

### 2.1. Materials

Tetraethyl orthosilicate (TEOS), anhydrous ethanol (ETOH), ammonia solution (NH_4_OH, 28% in H_2_O), tetrachloroauric acid (H[AuCl_4_]), sodium borohydride (Na[BH_4_]), 3-aminopropyltrimethoxysilane (APTES), and R6G were purchased from Shanghai Aladdin Biochemical Co., Ltd. (Shanghai, China). Hydrogen peroxide (H_2_O_2_), sulfuric acid (H_2_SO_4_), potassium carbonate (K_2_CO_3_), and formaldehyde (HCHO) were obtained from Guangzhou Chemical Reagent Factory (Guangzhou, China). Sodium citrate (Na-citrate) was supplied by Sinopharm Chemical Reagent Co., Ltd. (Shanghai, China). Deionized water (DI water; resistivity = 18.25 MΩ·cm) was prepared using the Milli-Q system. Si wafers with a thickness of 1 mm were purchased from Tebo Technology Co., Ltd. (Harbin, China). The Si wafers were continuously ultrasonic cleaned in ETOH and DI water and then immersed in piranha solution (H_2_SO_4_: H_2_O_2_-7:3, *v*/*v*) at 80 °C for 1 h. After being rinsed thoroughly with DI water, the Si wafers were dried in an oven at 80 °C for 1 h (i.e., until use). 

### 2.2. Preparation of Substrates

#### 2.2.1. Preparation of APTES-Functionalized SiO_2_ NPs

SiO_2_ NPs were prepared using a modified Stöber method, as shown below. Volumes of 42.5 mL of DI water, 32.5 mL of ETOH, and 19 mL of NH_4_OH were mixed under magnetic stirring in a 30 °C water bath environment for 30 min. Volumes of 90 mL of DI water and 8 mL of TEOS were mixed in the same environment and then added to the mixed solution; this operation needed to be completed evenly, and the process was slowed within 1 min. After magnetic stirring and heating for 2 h, the resulting solution was washed and redispersed five times using a solution (DI water: ETOH-1:1, *v*/*v*) and dried under vacuum at 60 °C for 3 h. Finally, we obtained white, solid SiO_2_ NPs.

A total of 0.2 g of SiO_2_ NPs was dissolved in 50 mL of ETOH via magnetic stirring at room temperature for 30 min. A total of 1 mL of APTES was dissolved in 10 mL of ETOH using the same approach. Then, the two solutions were mixed evenly via magnetic stirring. The mixed solution was then heated in an 80 °C water bath and refluxed for 2 h. After cooling to room temperature, the SiO_2_ NP solution was purified via differential centrifugation and redispersion five times. Finally, the precipitates were redispersed in 9.8 mL of DI water to form a 2 wt% APTES-functionalized SiO_2_ NP suspension, which was stored in a 4 °C refrigerator for spin coating.

#### 2.2.2. Preparation of Gold Seed Solution

Gold seeds with an average size of 7.17 nm were synthesized using the following process. Volumes of 10 μL of 1 wt% H[AuCL_4_] aqueous solution and 100 μL of 1 wt% Na-citrate aqueous solution were diluted with DI water until reaching 10 mL. A total of 500 μL of 10 mM Na[BH_4_] aqueous solution was then added as a reductant. The mixed solution was stirred for 30 min in an ice bath environment. Finally, the gold seed solution turned pink and was stored in a 4 °C refrigerator until use.

#### 2.2.3. Preparation of Gold Growth Solution

Volumes of 50 mg of K_2_CO_3_ and 5 mL of 1 wt% H[AuCL_4_] aqueous solution were added to 95 mL of DI water in a brown reagent bottle and placed in a dark environment for 12 h to form a gold growth solution which was used for a secondary reduction to form the gold shell of the SiO_2_@Au NPs.

#### 2.2.4. SERS Substrate Preparation

A Au-SiO_2_ NP substrate was prepared using the vertical deposition method, as shown in Figure 1a, which is a type of conventional doping method. A total of 5 mL of gold seed solution was added to 25 mL of ETOH solution of 3 wt% SiO_2_ NPs; subsequently, the solution was stirred for 30 min and then sonicated for 1 h so that it was mixed evenly. A Si wafer was vertically inserted into the solution, fixed, and then transferred to a vacuum-drying oven heated to 55 °C for 3 h. After the solution had evaporated, a Au-SiO_2_ NP photonic crystal array formed on the Si wafer.

A SiO_2_@Au NP substrate was prepared via spin-coating and in situ reduction methods, as shown in Figure 1b. First, a Si wafer was cut into squares with sides of 10 mm, and the APTES-functionalized SiO_2_ NP solution was transferred to the Si wafer using the spin-coating method to form a photonic crystal array. The Si wafer was cured in an oven at 60 °C for 1 h. Second, the Si wafer was soaked in 10 mL of gold seed solution at 4 °C, at which point tiny gold NPs coupled with the APTES-functionalized SiO_2_ NPs via their amino groups. After 6 h, the Si wafer was removed from the gold seed solution and rinsed with DI water five times to remove any free gold seeds. Then, the Si wafer was placed in gold growth solution in a dark environment, and 50 μL of HCHO was added as a reducing agent. A gold shell was grown on the exposed gold nucleation sites on the surface of the SiO_2_ NPs. Twelve hours later, the Si wafer was removed and rinsed with DI water. Finally, we obtained a SiO_2_@Au NP array on the Si wafer.

#### 2.2.5. Characterization

SEM images were captured using a field emission scanning electron microscope (EFSEM, model Gemini SEM 500). TEM images were obtained using transmission electron microscopy (TEM, HT7800). A Renishaw confocal Raman spectrometer with a 633 nm laser excitation source, a spot size of 1.03 μm, and a 50× objective lens was used to record the SERS signals of the Au-SiO_2_ NP and SiO_2_@Au NP substrates. An absorbance spectrum was acquired using an ultraviolet spectrophotometer and collected with a range of 300 nm to 800 nm. In this study, we conducted an experiment consisting of typical SERS liquid sample detection [21]. R6G was employed as the probe molecule for SERS.

## 3. Results and Discussion

As shown in Figure 2a, the TEM image of the SiO_2_ NPs shows that their surfaces were smooth and spherical. With an average diameter of 302.59 nm, the size distribution of the SiO_2_ NPs is Gaussian. After functionalization via APTES, the amino groups on the surface of the silica NPs became positively charged, while the gold seeds were electronegative, so they were easily coupled through electrostatic interactions [26], which provided a greater coupling strength of the two materials than that afforded using the doping methods. Figure 2c shows an SEM image of the APTES-functionalized SiO_2_ NP array obtained via spin coating, with a coating rate = 1000 rpm, time = 30 s, and volume = 40 μL. The regular arrangement of the photonic crystal structure covered the whole substrate and formed an opal structure arrangement; few vacancies or clusters were observed. According to previous reports, the average SERS intensity of this opal nanostructure is four orders of magnitude higher than that of a single nanostructure [27]. Figure 2d shows a TEM image of the gold seeds with an average size of 7.17 nm, which provided a good LSPR effect for the substrate. The absorbance spectrum of the gold seed solution shown in Figure 2e suggests that there was a typical peak at 522 nm, which was caused by the excitation of the surface plasmon vibrations of the Au NPs [28]. Figure 2f reveals the morphology of the Au-SiO_2_ NP array prepared using the doping method. There are only a few Au NPs attached to the SiO_2_ photonic crystal array, which limited the SERS performance of the substrate. As shown in Figure 2g, compared with the SiO_2_ NP in Figure 2a, the SiO_2_@Au NP array has plenty of Au seeds on its surface; these Au NPs coupled with the amino groups served as “seeds” for the growth of the gold shells, and the SiO_2_ spheres were not completely covered by the gold shells, resulting in rough surfaces and some “hot spots” on the surfaces of the SiO_2_ NPs. These special core–shell structures contributed to the enhancement of the Raman signal. Figure 2h is an SEM image of the morphology of the SiO_2_@Au NP array. The SiO_2_@Au NP array retained its opal structure, and the same is true for the SiO_2_ NP array in Figure 2c. Compared with the Au-SiO_2_ NP array presented in Figure 2f, the surface of the SiO_2_@Au NP array was evenly loaded with more Au NPs, leading to better SERS properties. In the larger-scale SEM image shown in Figure 2i, it can be clearly seen that at the scale of 10 μm, the SiO_2_@Au NPs still maintain a regular arrangement. It can be observed that immersion in the gold seed and gold growth solutions did not destroy the photonic crystal structure of the substrate.

The usual method for dealing with photonic crystals with an opal structure, also known as a face-center cubic structure, involves a modified version of Bragg’s law [29]:(1)$$\lambda_{max}=2d_{hkl}\sqrt{n_{eff}^{2}-{sin}_{\theta }^{2}}$$
(2)$$n_{eff}^{2}=fn_{0}^{2}+\left(1-f\right)n_{c}^{2}$$

When h k, and l are all odd, the crystal face spacing of the two adjacent layers of SiO_2_ NPs in the direction of incident light can be represented as follows:(3)$$d_{hkl}=\frac{\sqrt{2}D}{\sqrt{h^{2}+k^{2}+l^{2}}}$$
where $\lambda_{max}$ is the wavelength of the maximum reflection intensity of the SiO_2_@Au opal structure, $n_{eff}$ is the effective refractive index of the SiO_2_ sphere photonic crystal, and θ is the incident angle with respect to normal incidence. Vertical incident light was used in the test, so the incidence θ=0°. In a face-center cube, d111=23D, where D = 302.59 nm is the diameter of the SiO_2_ sphere. The filling coefficient of the SiO_2_ microsphere is f=0.7. $n_{0}=1.45$ is the refractive index of the SiO_2_ sphere, and $n_{c}=1$ is the refractive index of air. Based on (1)–(3), we determined that the wavelength of the maximum reflection of the SiO_2_@Au opal structure is 667 nm. During the interaction between incident light and a photonic crystal, when the wavelength of the incident light coincides with the photonic band gap of the photonic crystal, the reflection intensity reaches its maximum. So, the wavelength of the maximum reflection intensity of the SiO_2_@Au opal structure is where the photonic band gap lies. Therefore, the photonic band gap of the SiO_2_@Au opal structure is at around 667 nm. According to a previous report [30], SERS enhancement is strongly dependent on the spectral alignment between the position of the PBG and the Raman laser. In this study, the wavelength of the Raman laser used was 633 nm, which is basically consistent with the 667 nm photonic band gap of the SiO_2_@Au opal structure. In this case, the PBG of the SiO_2_@Au opal structure inhibits the Raman scattering photons transmitting through the structure in the incident direction. Moreover, the PBG of the SiO_2_@Au opal structure leads to the interference of backward-reflected Raman-scattered light, resulting in a strong enhancement of the backward reflection Raman scattering signal [25].

We captured energy-dispersive X-ray (EDX) images to confirm the chemical composition of the SiO_2_@Au NPs. The results presented in Figure 3 indicate that the SiO_2_ was covered by Au, and the distribution of Au is roughly the same as that of N. This result confirms that the Au NPs grew on the surface of the SiO_2_ at the amino groups. Such a distribution of Au NPs created a high-density LSPR effect and “hot spots”, both of which greatly contributed to SERS enhancement. The characteristic peak of Al in the EDS image is due to the presence of Al in the instrument used.

Different concentrations of R6G solution ranging from 1 × 10^−4^ M to 1 × 10^−9^ M were used to study the SERS sensitivity of the Au-SiO_2_ NP and SiO_2_@Au NP substrates. The results in Figure 4a correspond to the Au-SiO_2_ NP substrate; they show that vibration peaks at 612, 773, 1180, 1312, 1362, 1509, and 1649 cm^−1^ were observed in the spectra of the R6G molecules. The results in Figure 4b corresponding to the SiO_2_@Au NP substrate show similar vibration peaks at 612, 771, 1180, 1311, 1362, 1510, and 1650 cm^−1^. The attributions of these peaks are shown in Table 1. Compared with the previous report presented in [31], some peaks were slightly shifted in the SERS spectra; these changes were due to the effects of the different substrates. For the Au-SiO_2_ NP substrate, the above characteristic peaks are still clearly visible when the concentration of R6G was as low as 1 × 10^−7^ M, while the detection limit of the SiO_2_@Au NP substrate was 1 × 10^−8^ M. Figure 5 shows the SERS spectra of the two kinds of substrates tested with respect to 1 × 10^−5^ M R6G; on average, the Raman intensity of the SiO_2_@Au NP substrate was 27.08% higher than that of the Au-SiO_2_ NP substrate. The SERS spectra indicate that the SiO_2_@Au NP substrate had higher SERS intensity than the Au-SiO_2_ NP substrate.

To determine the reproducibility of the substrate, we prepared 10 pieces of each of these two substrates using the same process as that presented in 2.2.4, for which each piece was prepared independently. A 1 × 10^−5^ M R6G solution was employed to evaluate the performance of each substrate. As shown in Figure 6, the SERS spectra of the two kinds of 10 substrates were roughly the same, and all the vibration peaks mentioned above could be observed. In order to analyze the reproducibility of these two kinds of substrates, we selected the Raman intensities of four vibration peaks (shown in Figure 7), whose RSDs were calculated and presented in Table 2. The average RSD of the Au-SiO_2_ NP substrates was 19.49%; while this is acceptable according to similar work [32], it pales in comparison to the average RSD of 9.82% of the SiO_2_@Au NP substrate.

To further study the stability of the SiO_2_@Au NP substrate, we analyzed the SERS spectra of the 1 × 10^−5^ M R6G solution acquired from a freshly prepared substrate and a substrate preserved for one month, as shown in Figure 8. The SERS spectra of the SiO_2_@Au NP substrate did not change significantly after one month of storage, and all the vibration peaks mentioned above could still be observed, indicating that the substrate had good stability with respect to the Raman signal.

### Enhancement Factor

The Raman enhancement factor (EF) is an important criterion for evaluating SERS substrate performance. The EF value in this study was evaluated according to the following standard equation:(4)$$EF=\frac{I_{SERS}}{I_{CRS}}\times \frac{N_{CRS}}{N_{SERS}}$$

Here, $I_{SERS}$ is the integrated intensity of the absorbed molecules of the 10^−5^ M R6G solution on the SERS substrate; $I_{CRS}$ is the integrated intensity of the same vibrational band according to conventional Raman spectroscopy; and $N_{SERS}$ and $N_{CRS}$ are the numbers of R6G molecules illuminated by the laser’s focusing point under SERS (SERS substrate) and conventional Raman spectroscopy (Si wafer) measurement environments. In this study, the diameter of the laser beam focused on the samples (D) and its penetration depth into the samples (P) were determined using the following equation:(5)$$D=\frac{1.22\lambda_{laser}}{N_{A}}$$
(6)$$P=\frac{2.2n\lambda_{laser}}{\pi N_{A}^{2}}$$
where $\lambda_{laser}\;$ = 633 nm is the wavelength of the Raman laser, n=1.5930 is the refractive index of solid R6G powder, and $N_{A}=0.75$ is the numerical aperture. D was calculated to be 1.03 μm, and P was calculated to be 1.26 μm.

The illuminated volume (V) is determined by D and P and can be calculated using the following equation:(7)$$V=P\times A=P\times \pi {\left(\frac{D}{2}\right)}^{2}$$

The calculated value of V is 1.05 μm^3^; then, $N_{CRS}$ can be determined as follows:(8)$$N_{CRS}=N_{A}\times \frac{\rho V}{M_{r}}$$
where $\rho =1.26\;g/{cm}^{3}$ is the density of R6G, and $M_{r}=479\;g/mol$ is the molar mass of R6G. The value of $N_{CRS}$ was calculated to be approximately 2.072 × 10^−15^.

$N_{SERS}$ is determined by the area illuminated by the Raman laser on the SERS substrates. The value of $N_{SERS}$ can be obtained as follows:(9)$$N_{SERS}=N_{A}\times \frac{AC_{SERS}V_{1}}{A_{1}}$$

We assumed that $V_{1}$ = 20 μL of 1 × 10^−5^ M R6G solution was uniformly distributed on the surface of the SERS substrates. According to the size of the Si wafer substrate, it was determined that the SERS substrate area immersed in 20 μL of the R6G solution ($A_{1}$) was equal to 1 cm^2^. $C_{SERS}$ = 1 × 10^−5^ M is the concentration of R6G solution. The value of $N_{SERS}$ was calculated to be approximately 1.249 × 10^−18^.

According to Figure 5, the $I_{SERS}$ values of the SiO_2_@Au NP and Au-SiO_2_ NP substrates are 17,539.32 and 12,290.75 at 1180 cm^−1^. Additionally, $I_{CRS}$ is 19.27 at 1180 cm^−1^. Hence, the EF values of the SiO_2_@Au NP and Au-SiO_2_ NP substrates were calculated to be approximately 1.51 × 10^6^ and 1.06 × 10^6^ at 1180 cm^−1^, respectively.

Table 3 shows the SERS properties of the Au-SiO_2_ NP and SiO_2_@Au NP substrates. Although the Au-SiO_2_ NP substrate presented good performance, the SiO_2_@Au NP substrate outperformed the Au-SiO_2_ NP substrate in every area listed in the table. Considering that they are both composed of the same materials, the improvement in SERS properties can be attributed to the optimization of the preparation method.

## 4. Conclusions

In summary, we have described a new method, which differs from the conventional doping method, for the fabrication of a photonic-crystal-structured SERS substrate. Based on the SiO_2_@Au NP array, the coupling strength between the Au NPs and the SiO_2_ spheres was improved via surface modification, which was reflected in the improved reproducibility of the substrate. The SiO_2_@Au photonic crystal structure also contributed to the SERS enhancement. The loading capacity of Au was increased using in situ reduction methods, and the evenly distributed gold shell provided numerous “hot spots” and a stronger LSPR effect for the substrate. Using R6G as a probe molecule, we studied the SERS characteristics of the substrate. Compared with the substrate prepared using a doping method (namely, the Au-SiO_2_ NP substrate), the SiO_2_@Au NP substrate not only improved the detection limit by an order of magnitude equal to 1 × 10^−8^ M but also improved its RSD of reproducibility from 19.49% to 9.82%; the value of the enhancement factor also improved from 1.06 × 10^6^ to 1.51 × 10^6^. Moreover, the SiO_2_@Au NP substrate also reproduced the SERS spectra after one month. These improvements indicate that our SiO_2_@Au NP substrate has potential applications in materials science, bioscience, and analytical science.

## Figures and Tables

**Figure 1 nanomaterials-13-02156-f001:**
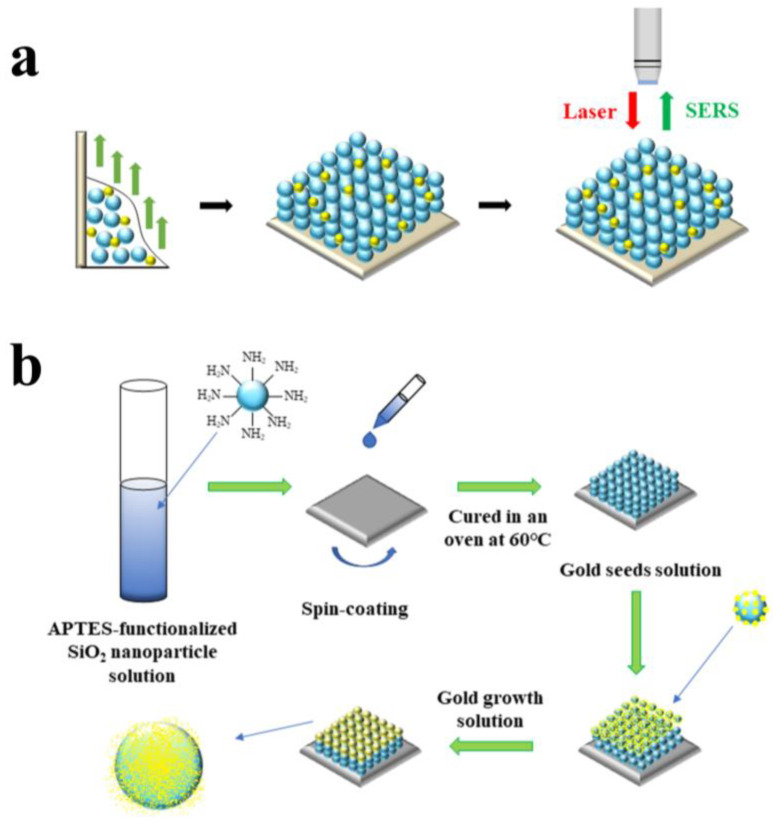
Schematic diagram of the SERS substrate preparation procedure: (**a**) preparation of Au-SiO_2_ NP substrate; (**b**) preparation of SiO_2_@Au NP substrate.

**Figure 2 nanomaterials-13-02156-f002:**
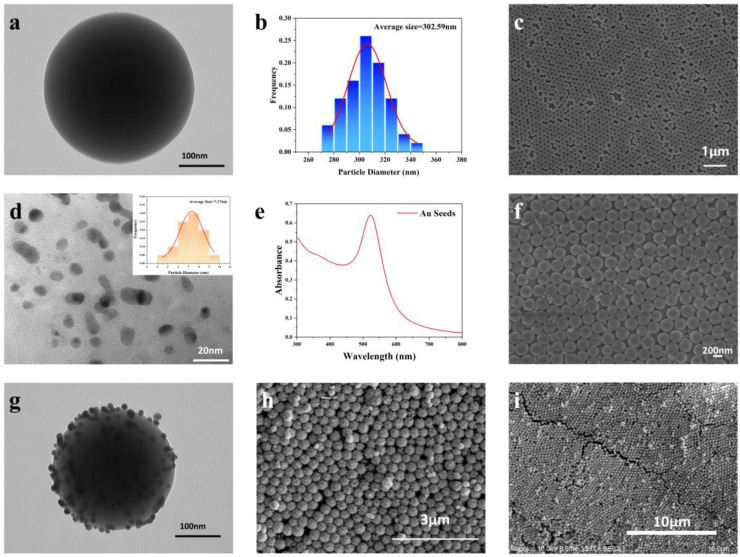
(**a**,**b**) TEM images of APTES-functionalized SiO_2_ NPs with an average size of 302.59 nm; (**c**) SEM image of APTES-functionalized SiO_2_ NPs array; (**d**) TEM image of gold seeds with an average size of 7.17 nm; (**e**) absorbance spectrum of gold seed solution; (**f**) SEM image of Au-SiO_2_ NP substrate; (**g**) TEM image of SiO_2_@Au NP; (**h**,**i**) SEM images of SiO_2_@Au NP substrates.

**Figure 3 nanomaterials-13-02156-f003:**
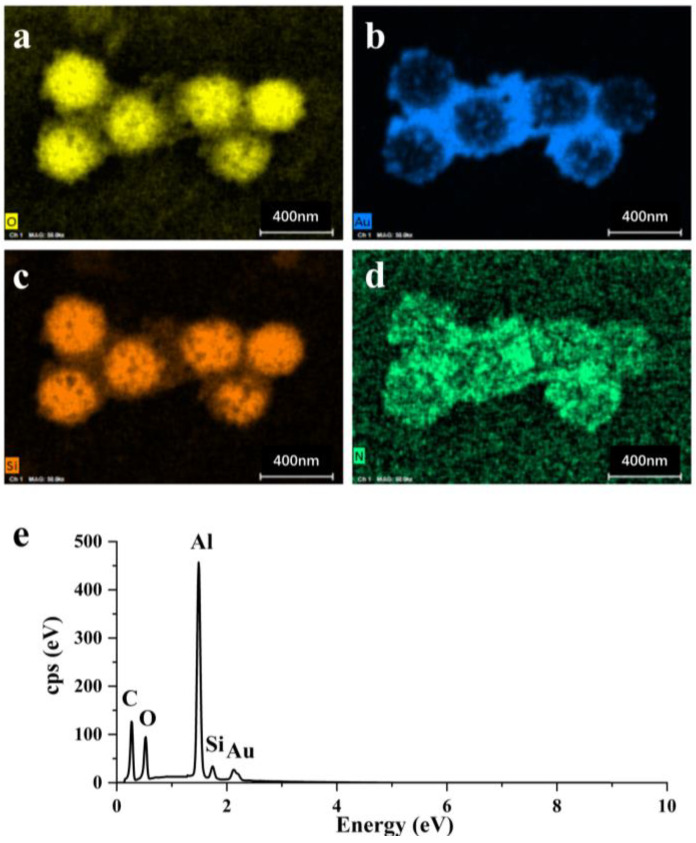
The EDX images of elements in the SiO_2_@Au NP array: (**a**) O; (**b**) Au; (**c**) Si; (**d**) N. (**e**) EDS spectrum of SiO_2_@Au NP.

**Figure 4 nanomaterials-13-02156-f004:**
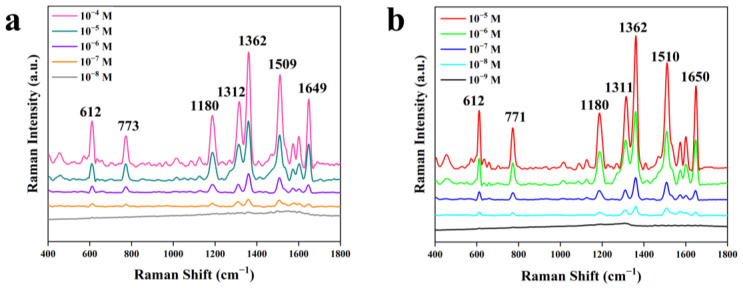
SERS spectra for R6G adsorbed on substrates at different concentrations: (**a**) Au-SiO_2_; (**b**) SiO_2_@Au.

**Figure 5 nanomaterials-13-02156-f005:**
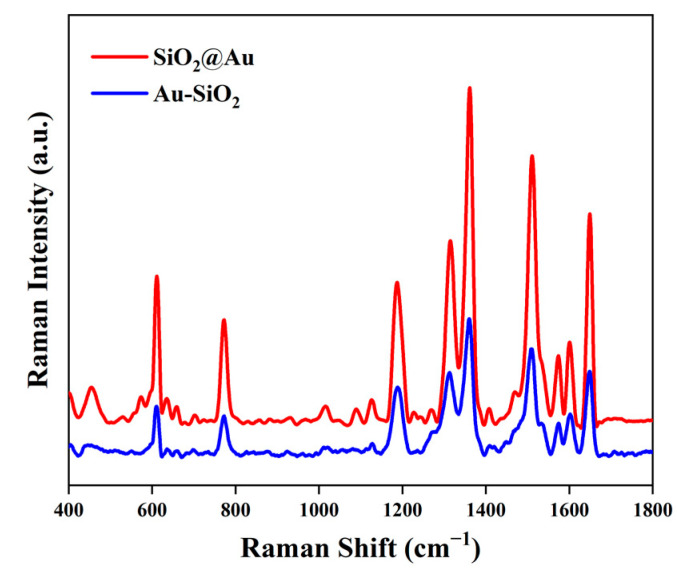
SERS spectra of 1 × 10^−5^ M R6G adsorbed on Au-SiO_2_ NP substrate and SiO_2_@Au NP substrate.

**Figure 6 nanomaterials-13-02156-f006:**
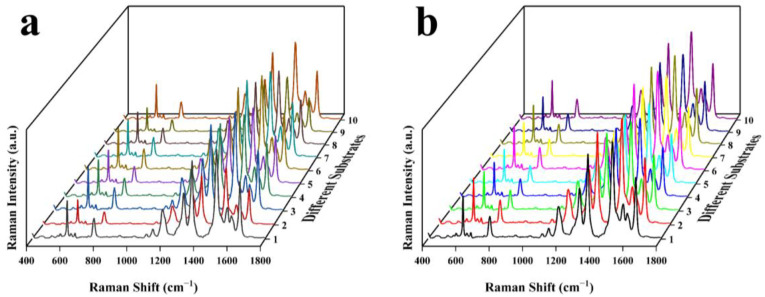
SERS spectra of 1× 10^−5^ M R6G adsorbed on 10 different substrates: (**a**) Au-SiO_2_ NP substrate; (**b**) SiO_2_@Au NP substrate.

**Figure 7 nanomaterials-13-02156-f007:**
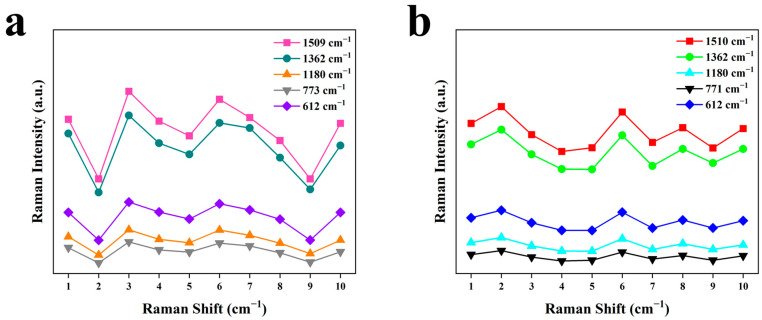
Raman intensities of five different vibration peaks on 10 different substrates: (**a**) Au-SiO_2_ NP substrate; (**b**) SiO_2_@Au NP substrate.

**Figure 8 nanomaterials-13-02156-f008:**
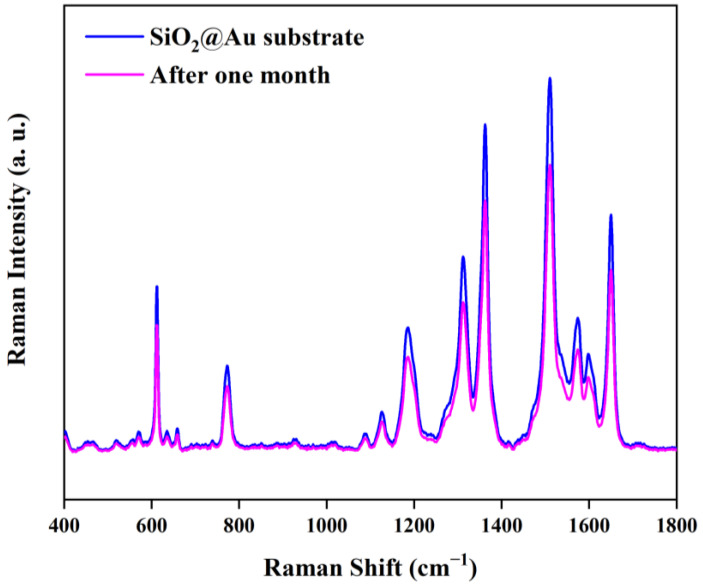
SERS spectra of 1 × 10^−5^ M R6G solution acquired from a freshly prepared substrate and a substrate preserved for one month.

**Table 1 nanomaterials-13-02156-t001:** Comparison and description of vibration peaks in SERS spectra.

Au-SiO_2_	SiO_2_@Au	Reported Result [31]	Vibration Assignments
612	612	614	C-C-C ring ip bend
773	771	773	C-H op bend
1180	1180	1181	C-H ip bend
1312	1311	1310	Arom C-C str
1362	1362	1363	Arom C-C str
1509	1510	1509	Arom C-C str
1649	1650	1650	Arom C-C str

**Table 2 nanomaterials-13-02156-t002:** RSDs of five different vibration peaks.

Substrate	612	773/771	1180	1362	1509/1510	Average
Au-SiO_2_	18.60%	18.86%	17.82%	18.61%	18.55%	19.49%
SiO_2_@Au	10.06%	9.50%	10.07%	9.91%	9.56%	9.82%

**Table 3 nanomaterials-13-02156-t003:** SERS properties of the two kinds of substrates.

Substrate	Detection Limit	Average RSD of Reproducibility	EF Value
Au-SiO_2_	1 × 10^−7^ M	19.49%	1.51 × 10^6^
SiO_2_@Au	1 × 10^−8^ M	9.82%	1.06 × 10^6^

## Data Availability

The raw data are not publicly available at this time but may be obtained from the authors upon reasonable request.

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
