# Peer review of "Preparation of SiO2@Au Nanoparticle Photonic Crystal Array as Surface-Enhanced Raman Scattering (SERS) Substrate"

_nanomaterials, 2023, doi:10.3390/nano13152156_

Round 1

Reviewer 1 Report

There are some questions and corrections that should be taken into consideration before publication:

1. Line 37 «an increase the SERS activity up to 10-8». The sentence is unclear. What is the 10-8? Concentration (M, mol/l)? Concentration of which probe? And if authors speak about the probe concentration than sensitivity instead of activity should be used.

2.  Line 70. I do not find the explanation for the abbreviation of «R6G». It should be clarified in the paper.

3. Line 95. The same question about APTES. Both terms can be well-known for the scientists working in the narrow field but the paper is devoted for the wide auditory and abbreviations should be clarified.

4. Line 89. The unit dimension for the water resistivity is «MΩ•cm»

5. line 174 «shown at Figure 2i, it can be clearly seen that at the scale of 10nm». As I see the figure 2i has a scale of 10 mkm but not 10 nm.

6. Lines 178-193. The plasmon absorption is not the «diffraction peak». Moreover, it is absolutely not obvious that plasmon absorption is some-how interrelated with the X-ray diffraction (interplane distances d). Certainly, that plasmon absorption wavelength depends on the size of the particle, but the formula (1) and (3) need explanation.  But why d111, why Bragg equation (1) is suitable for the wavelength calculation?  The theory must be clearly explained or the reference to corresponding papers should be given.

7. Line 223. The reproducibility? The samples used for the test were prepared independently from the very beginning or the samples were prepared from one finally modified SiO2 wafer which was cut to pieces. Better to clarify that question.

Reviewer 2 Report

The manuscript contains fruitful results about the preparation of SiO2@Au nanoparticles as a SERS substrate.  The fabrication method resulted in increased Au loading capacity compared to the traditional doping method.  The detection limit and RSD of reproducibility were increased at an important rate. 

Some minor mistakes should be revised e.g.:

line 84, line 110, line 116. HAuCl4 is a complex compound, it had better written as H[AuCl4]. 

line 84, line 98. NH3.H2O is not an existing compound, that can be written as NH4OH. It is not the hydrate of ammonia, but ammonium hydroxide.

line 84, line 112 NaBH4 is a complex compound, it had better write as Na[BH4]. 

line 85. Not need capital letters for 3-Amino... 

Fig.2b is not a TEM image but a distribution curve. 

Table 1 (line 222), Description, does not need capital D. 

Reviewer 3 Report

Song et al., fabricated a surface-enhanced Raman scattering (SERS) substrate utilizing a photonic crystal array composed of a SiO2@Au core-shell structure. The efficacy of surface-enhanced Raman scattering (SERS) was confirmed through the use of Rhodamine 6G (R6G). The SiO2@Au NP array exhibited superior SERS activity and reproducibility, as well as the ability to reproduce SERS spectra even after a one-month period. The detection limit of Rhodamine 6G (R6G) on a silicon dioxide-coated gold nanoparticle (SiO2@Au NP) array was determined to be 1×10-8 M (mol/L), and the reproducibility of the measurements was assessed with a relative standard deviation of 9.82%. However, further revisions are required for this work to be accepted for publication.

1.       The significance of the size of SiO2 nanoparticles and their SERS efficacy has been addressed by the author in the discussion section.

2.       If feasible, the author should incorporate the COMSOL calculation of the local field enhancement of the SERS substrate.

3.       The author has included an analysis of the improvement in SERS substrate performance and has conducted a comparative evaluation with other SERS substrates.

4.       Additionally, it is suggested that the author incorporates an enhancement comparison table into the text.

5.       In the discussion section, the author includes and examines the SERS mechanism of the current substrate.

Reviewer 4 Report

The work is important from the point of view of improving the sensitivity of the SERS method. It should be written more concisely with emphasis on the most important things. So I have a few comments:

Line 34-43: These theories have been known for many years and it is not necessary to quote them, especially since it does not bring anything new

Line 52: improve sentence order

Line 61: improve sentence order

Line 67: are you sure  2.2x10-7?

Line 71: express it more clearly e.g. “we propose a method that is based on ..... which differs from the classic method based on ....            The aim of the research undertaken needs to be more clearly and definitely emphasized. What do we really expect.

Line 77: how was this 9.82% calculated?

The caption under Fig.1 is very short - it needs to be expanded. Expand the abbreviation APTES.

Do we get a functionalized SiO2 nanoparticle at the end of the preparation shown in Fig.1b?

Line 117: How was the nanoparticle size measured with such accuracy?

Fig.2b is not a TEM image but rather a grain size distribution

The insert in Fig.2d is completely illegible

Line 165-166: incomprehensible

Line 178-192: Where did the diffraction considerations come from ?. No results shown. Besides, what's that supposed to explain?

The focus should be on Fig.2a and c and Fig.2g, h and i. Comparison and highlight differences better.

Should be reviewed by a native speaker in terms of word order

Round 2

Reviewer 1 Report

The authors take into consideration all my remarks. The paper can be published.